



# Moisture sources contribute to precipitation change in the Three Gorges Reservoir Region during 1979–2015

Ying Li[1], Chenghao Wang[2], Hui Peng[1], Shangbin Xiao[1], Denghua Yan[3]

[1]College of Hydraulic & Environmental Engineering, China Three Gorges University, Yichang, 443002, China
[2]Department of Earth System Science, Stanford University, Stanford, CA 94305, USA
[3]State Key Laboratory of Simulation and Regulation of Water Cycle in River Basin, Water Resources Department, China Institute of Water Resources and Hydropower Research (IWHR), Beijing, 100038, China

*Correspondence to*: Denghua Yan (yandh@iwhr.com)

**Abstract.** Precipitation change in the Three Gorges Reservoir Region (TGRR) plays a critical role in the operation and
regulation of the Three Gorges Dam (TGD) as well as the protection of residents and properties. The potential impacts of the TGD on local and regional circulation patterns, especially the precipitation patterns, have received considerable attention since its construction. However, how the moisture transport affects the precipitation change in the TGRR spatially and temporally remains obscure. In this study, we investigate the long-term moisture sources of precipitation as well as their contributions to the precipitation change over the TGRR using an atmospheric moisture tracking model. Results suggest that
although with seasonal variation, the moisture contributing to the TGRR precipitation primarily originate from the areas southwest of the TGRR dominated by the Indian summer monsoon. In particular, the sources with the highest annual moisture contribution are the southwestern part of the Yangtze River Basin and the southeastern tip of the Tibetan Plateau (TP). On average, 41%, 56%, and 3% of the TGRR precipitation originate from ocean, land, and local recycling, respectively. In addition, the decreased precipitation over the TGRR during 1979–2015 is mainly attributed to the significantly decreased
moisture contribution from the source regions southwest of the TGRR (especially around the southeastern tip of the TP). Compared to dry years, the higher precipitation in the TGRR during wet years is contributed by the extra moisture from the southwestern source regions delivered by the intensified southwesterly monsoon winds.

## 1 Introduction

Dams and reservoirs are of particular importance to global human development via water supply, flood control, drought
mitigation, and electricity generation. As the world's largest hydropower project to date, the Three Gorges Project has received considerable attention since the construction of the Three Gorges Dam (TGD) in 1993. In particular, extensive studies have evaluated the potential environmental impacts and the local and regional climate changes in the Three Gorges Reservoir Region (TGRR) induced by this project, especially the potential consequences of dramatic land use and land cover change (*Morgan et al.*, 2012; *Li et al.*, 2013; *Xu et al.*, 2013). Located at the eastern edge of the Sichuan Basin between
Chongqing municipality and Yichang city in Hubei province, the TGRR covers over 58,000 km[2] of the Yangtze River Basin



(YRB) with the reservoir stretching about 660 km (*Li et al.*, 2013) (Fig. 1). The climate of this region is influenced by two major monsoon systems, East Asia monsoon and Indian summer monsoon, and is characterized by wet summers and dry winters with over 75% of the precipitation in June–September (*Xu et al.*, 2004; *Wei et al.*, 2012; *Wang et al.*, 2018; *Fremme and Sodemann*, 2019). The rapid expansion of water surface due to the reservoir impoundment can potentially alter the

vertical profiles of temperature and humidity as well as the exchange of heat and moisture in the lower atmosphere, leading to changes in patterns of precipitation and evaporation (*Miller et al.*, 2005; *Hossain et al.*, 2009; *Biemans et al.*, 2011; *Yigzaw et al.*, 2013; *Li et al.*, 2019b). The accelerating hydrological cycle and increasing frequencies of extreme events caused by global warming in recent decades (*IPCC*, 2014; *Pihl et al.*, 2019) further amplify the uncertainties of precipitation in this region. A thorough investigation of the long-term precipitation changes in the TGRR, in particular their driving

mechanisms and the potential local and regional impacts of the dam construction, is therefore imperative.

Existing studies have shown spatial and temporal changes in precipitation characteristics in the TGRR during recent decades, primarily based on ground observations. For example, *Xiao et al.* (2010) found a dipole pattern of precipitation change in the surrounding areas of the Three Gorges Reservoir (TGR) from 1960 to 2005, i.e., increased precipitation in the north but

decreased in the south. Another study (1958–2007) revealed the interdecadal changes in TGRR precipitation with the wettest period being the early 1980s and a relatively dry period after 1990s (*Zhao et al.*, 2010). *Zhao et al.* (2010) also observed significant increases in precipitation intensity over the same five decades. On the other hand, the precipitation changes in the TGRR as induced by TGD are mainly investigated through station measurements, satellite datasets, and numerical simulations. For instance, *Lü et al.* (2018) identified an increasing trend of extreme precipitation during 1959–2013 in the

TGRR based on weather station data and the Tropical Rainfall Measuring Mission (TRMM) dataset. With the hypothesis of relatively local impacts, they concluded that the TGD has no detectable influence on extreme precipitation in the TGRR. In contrast, *Wu et al.* (2006) showed enhanced precipitation over the northwest TGRR but decreased precipitation in the vicinity of the TGD after the water level rose abruptly to 135 m in June 2003, based upon the TRMM dataset, the Moderate Resolution Imaging Spectroradiometer (MODIS) land surface temperature, and high-resolution mesoscale simulations. This

study also suggested that the climate effect of the TGD is on regional scale (~100 km) rather than on local scale (~10 km) as found in previous studies. In general, the long-term impact of the TGD on the TGRR precipitation is relatively weak, as suggested by both observational data and numerical simulations (*Miller et al.*, 2005; *Li et al.*, 2017; *Lü et al.*, 2018), and is likely dominated by the interannual (natural) climate variability (*Xiao et al.*, 2010; *Li et al.*, 2017). However, considering the dynamic interplay between local impacts and regional/global climate change, detecting the underlying drivers of TGRR

rainfall change based solely on local or regional precipitation data can be very challenging. It is necessary to take into account moisture sources at different spatial scales as well as local and nonlocal moisture transports that lead to precipitation.

Atmospheric circulations play an essential role in the global and regional hydrological cycle, which transport the moisture evaporated from one location and precipitate it out elsewhere (*Trenberth et al.*, 2011; *Gimeno et al.*, 2012; *Gimeno et al.*,





2020). In particular, the sources and transports of atmospheric moisture are regulated by climate and synoptic patterns, and their modification reflects the changes in land-atmosphere interactions, atmospheric circulation patterns, and more broadly, the regional hydrological cycle (*Van der Ent et al.*, 2010; *Gimeno et al.*, 2012). Previous studies have suggested that the major source regions of moisture for precipitation over the YRB are the South China Sea, the western Pacific, Bay of Bengal, Arabian Sea, South China, and Mainland Southeast Asia (*Wei et al.*, 2012; *Chen et al.*, 2013; *Wang et al.*, 2018; *Fremme and*

*Sodemann*, 2019). A previous study based on a Lagrangian particle dispersion model demonstrated that the interannual variability of the summer precipitation over the YRB is closely related to the moisture supply from the Bay of Bengal and the Arabian Sea (*Chen et al.*, 2013). However, *Wei et al.* (2012) pointed out that the oceanic contributions are mostly indirect due to the existence of continental recycling. This was later confirmed by *Fremme and Sodemann* (2019) who estimated the contribution of continental moisture sources to be 58.4%. In addition, *Xu et al.* (2004) defined a triangular region as the key

region of the moisture transport to the Meiyu rain belt in the YRB, which covers the Bay of Bengal, Mainland Southeast Asia, the southeastern Tibetan Plateau (TP), and part of the South China Sea. Nevertheless, these studies have relatively short study periods and are mainly focused on the entire YRB or the mid-lower reaches with simplified boundary of the river basin (usually a rectangular box). A comprehensive study that resolves the long-term moisture sources of the precipitation and how these sources have contributed to the precipitation change in the TGRR is hitherto absent.


This study aims to fill these research gaps based on comprehensive analyses for the period 1979–2015. More specifically, the main objectives are: (1) to numerically track the moisture sources of annual and seasonal precipitation over the TGRR; (2) to identify predominant sources that determine the interannual variability of precipitation in this region; (3) to detect the differences in moisture sources and transport pathways during wet and dry years. On the one hand, this study relies on the

long-term numerical moisture tracking, which more directly reveals the dynamics of source regions and transport pathways when compared to existing station-based and model-based analyses of precipitation and dam construction in the TGRR. On the other hand, different from previous moisture tracking studies over much greater regions (e.g., entire YRB), we focus exclusively on the TGRR, which largely reduces the potential noise induced by surrounding land and water bodies. Results of this study are except to shed new light on the determinants of hydrological change over the TGRR under climate change,

and provide insights into the effective water resource management, policy formulation, and disaster (e.g., flood and landslide) prevention and mitigation in the TGRR.

## 2 Method and data

### 2.1 Numerical atmospheric moisture tracking

In this study, we used an offline (a posteriori) Eulerian numerical atmospheric moisture tracking model, the Water

Accounting Model-2layers (WAM-2layers) (*Van der Ent et al.*, 2010; *Van der Ent*, 2014), to simulate the moisture transport and the source regions of moisture for the TGRR precipitation. Previous tests have suggested that the moisture tracking and





precipitation patterns simulated by WAM-2layers are nearly identical to those by a highly advanced online tracking model, but with a significant improvement in the computational speed (*Van der Ent et al.*, 2013). In addition, unlike some Lagrangian models such as the FLEXible PARTicle (FLEXPART) dispersion model and the Hybrid Single-Particle
Lagrangian Integrated Trajectory (HYSPLIT) model (*Stohl et al.*, 2005; *Stein et al.*, 2016), WAM-2layers diagnoses precipitation and evaporation separately (*Van der Ent et al.*, 2010). The WAM-2layers has been extensively used for regional and global moisture tracking in existing studies (*van der Ent and Savenije*, 2013; *Keys et al.*, 2014; *Zemp et al.*, 2014; *Duerinck et al.*, 2016; *Zhao et al.*, 2016; *Van der Ent and Tuinenburg*, 2017; *Zhang et al.*, 2017; *Li et al.*, 2019a).

The governing equation of WAM-2layers is the mass balance solved for the two well-chosen layers (top and bottom). For example, in the bottom layer:

$$\frac{\partial S_g}{\partial t} = \frac{\partial (S_g u)}{\partial x} + \frac{\partial (S_g v)}{\partial y} + E_g - P_g + \xi_g \pm F_{V,g} \qquad (1)$$

where the subscript $g$ denotes the tagged atmospheric moisture (all possible phases of water) (*Van der Ent*, 2014); $S_g$ is moisture storage in bottom layer; $t$ is time; $u$ is the wind component in zonal (x) direction; $v$ is the wind component in
meridional (y) direction; $E_g$ is the evaporation entering the bottom layer; $P_g$ is the precipitation removed from the bottom layer; $\xi_g$ is the residual term; and $F_{V,g}$ is the vertical moisture exchange between the two layers. Note that the evaporation only occurs in the bottom layer.

In the tracking process, we assume that the precipitation is immediately removed from the storage of each layer and is well-
mixed following *Van der Ent et al.* (2010). The model assumes that the vertical exchange to be a closure term in the equation, but the balance cannot always be fully closed due to the existence of residual $\xi_g$. Therefore, the closure is defined by a ratio of residuals between the two layers, which is proportional to moisture content in each layer, i.e., $\xi_{g,top}/S_{g,top} = \xi_{g,bottom}/S_{g,bottom}$. The subscripts denote the two layers in the model. In this context, $F_{V,g}$ can be described as:

$$F_{V,g} = \frac{S_{g,bottom}}{S_g} (\xi^*_{g,bottom} + \xi^*_{g,top}) - \xi^*_{g,bottom} \qquad (2)$$

where $S_g$ is the total atmospheric storage, and $\xi^*_{g,bottom}$ and $\xi^*_{g,top}$ are the residuals before vertical exchange is considered. Note that the use of two layers has been proved to adequately capture the division of the sheared wind systems in the atmosphere (*Van der Ent et al.*, 2013).

The accurate simulation of moisture transport for the TGRR precipitation requires sufficiently high resolutions (*Gößling and*
*Reick*, 2013; *Van der Ent et al.*, 2013). To increase numerical stability and reduce computational cost, we used a spatial resolution of 1° × 1° with a time step of 0.25 h (temporal resolution) in all runs. Seventeen vertical layers from surface to the top of the atmosphere were chosen from the forcing reanalysis product. Following a global-scale trial test (*Van der Ent et al.*, 2013), the layer around 813 hPa (the eighth layer in our selected ERA-Interim dataset) was set as the vertical separation



between the bottom and top layers with a standard surface pressure. A 30-day period was used to spin-up the model in the
tracking process, which ensures that the vast majority of the tagged moisture (~95%) falls back to the ground surface (*Van der Ent et al.*, 2010; *Zhang et al.*, 2017). Based on our preliminary tests, we selected a tracking domain (20°S–60°N and 0°E–140°E) that covers nearly all potential source regions of TGRR precipitation. To clearly display the moisture source, the spatial extent of all following maps is smaller than the tracking domain.

## 2.2 Data source

The ERA-Interim reanalysis dataset was initiated by the European Centre for Medium-Range Weather Forecasts (ECMWF) in 2006 (*Dee et al.*, 2011). This global dataset covers a period from 1979 with a spatial resolution of 0.75° × 0.75°. The dataset from 1979 to 2015 was used to drive the WAM-2layers. The forcing data of the WAM-2layers are 6-hourly specific humidity (17 vertical layers) and 6-hourly wind fields (17 vertical layers), 6-hourly surface pressure, 3-hourly precipitation, 3-hourly evaporation, 6-hourly total column moisture, and 6-hourly vertical integrated moisture fluxes. For comparison, we
also used the Japanese 55-year Reanalysis data (JRA-55) developed by the Japan Meteorological Agency (*Kobayashi et al.*, 2015) and the Modern-Era Retrospective analysis for Research and Applications version 2 data (MERRA-2) from the National Aeronautics and Space Administration (*Gelaro et al.*, 2017) to track the moisture using the WAM-2layers. In these three sets of simulations (based on ERA-Interim, JRA-55, and MERRA-2), all variables were resampled to 0.25 h from their original temporal resolutions and were bilinearly interpolated to 1°×1° grids for consistency.


Two sources of precipitation data are used in this study. The first one is a high-quality precipitation dataset interpolated from 2,400 meteorological stations over China (*Zhao et al.*, 2014). This dataset was initiated by China Meteorological Administration (CMA). It uses a thin plate spline interpolation method with three-dimensional scheme, in which the digital elevation is also introduced. This dataset is 0.5°×0.5° gridded with a coverage period from 1961 to present, and it can be
downloaded directly from the China Meteorological Data Service Center (CMDC). For comparison, we also retrieved remotely sensed precipitation data from the TRMM Multi-satellite Precipitation Analysis (TMPA) (*Huffman et al.*, 2007) . Note that the time span of the retrieved TMPA is 1998–2015.

## 3 Results

The spatial distribution of linear trends in annual precipitation estimated from CMA observations over the entire YRB is
shown in Fig. 2a. It is clear that compared to other parts of the YRB, the TGRR is dominated by significant decreasing annual precipitation ($p<0.05$, based on two-sided t-test) during 1979–2015. The decreasing trend over this region is further illustrated with the time series of annual precipitation in Fig. 2b. The linear trends estimated using ground observations (CMA), reanalysis product (ERA-Interim), and satellite observation (TMPA) are in general consistent: –4.08 mm/yr (1979–2015; $p<0.05$), –6.24 mm/yr (1979–2015; $p<0.05$), and –4.85 mm/yr (1998–2015), respectively. Based on moisture tracking,





we then investigate the potential mechanisms that drive the decreasing trend of TGRR precipitation from the following three aspects: moisture sources (Section 3.1), the trends of their contributions to TGRR precipitation (Section 3.2), and moisture contributions during wet and dry years (Section 3.3).

## 3.1 Moisture sources of annual and seasonal precipitation in the TGRR

Figure 3 shows the contribution of the moisture (mm) from different source regions to the annual precipitation in the TGRR
averaged during 1979–2015, overlaid with the patterns of vertical integrated moisture fluxes. As simulated in WAM-2layers, moisture evaporated from the shaded source regions is carried by circulations and eventually contributes to the annual precipitation in the TGRR. Moisture sources with the highest contribution (>30 mm/yr) are mainly distributed in the regions southwest of the TGRR, primarily because relatively little precipitation is produced during the transport. Mainly along the plenty of moisture transport from the southwest, major source regions also include South China, southeastern TP, Mainland
Southeast Asia, part of the South China Sea, Bay of Bengal, part of Indian Subcontinent, and Arabian Sea. Although oceans evaporate more water than the land, moisture from oceans is largely lost by precipitation in the transport over the land. As a result, the contribution of the oceanic moisture source to the TGRR precipitation is relatively low (<6 mm/yr). Compared to the southwestern source regions, the southeastern source regions (e.g., East China and East China Sea) is more limited. This difference may largely result from the interaction between the Indian monsoon and the East Asia monsoon systems, while
the Indian monsoon tends to dominate the moisture transport toward the TGRR at the annual scale.

The seasonal variations of moisture sources contributing to the TGRR precipitation are illustrated in Fig. 4. In spring (March, April, and May), the moisture is mainly from regions west of the TGRR. In particular, the southeastern tip of the TP is one of the regions with the largest moisture contributions. The source regions further stretch west along the southern edge of the
TP and even cover part of the northern Arabian Sea. The dominant circulation system brings moisture to the TGRR in spring is the southern branch of the middle-latitude westerlies, which forms when westerlies hits the TP (a physical barrier). In summer (June, July, and August), when the Indian monsoon and East Asia monsoon start and the land gradually warms and wets, the major moisture source regions shift southwards, spanning across the entire Bay of Bengal and part of the Arabian Sea. In addition, the Indian summer monsoon (rather than the East Asia monsoon) dominates the transport of oceanic
moisture to the TGRR, as suggested by the strong moisture fluxes in Fig. 4. In autumn (September, October, and November), the moisture delivered from the areas southwest of the TGRR is remarkably reduced with the withdrawal of the Indian monsoon, and the moisture contribution from oceans largely decreases when compared to summer. Meanwhile, the moisture from regions southeast of the TGRR slightly increases, owing to the weakened southwestern wind. In winter (December, January, and February), the source regions largely shrink and only cover a small portion of the domain (mainly around the
southeastern edge of the TP). This pattern is in general consistent with that in spring (dominated by westerlies), although with much weaker moisture fluxes and smaller source regions. In general, the seasonal variations of source regions for the TGRR precipitation are influenced by the interactions among Indian monsoon system, East Asia monsoon system, and



middle-latitude westerlies, in which the regions southwest of the TGRR (as dominated by the Indian monsoon) play a leading role.


We then split the domain into two parts using 108°E (roughly across the center of the TGRR; see the dotted line in Fig. 3), i.e., the western part (WP) and the eastern part (EP) to further quantify the moisture contribution of different regions. In general, such division can roughly separate the contributions of two monsoon systems. The WP mainly represents source regions controlled by the Indian monsoon (in summer and autumn) and the westerlies (in spring and winter), whereas the EP

contains source regions influenced by the East Asia monsoon. The contribution of WP and EP to the monthly precipitation in the TGRR is shown in Fig. 5a. On average, about 80% of the total moisture is from the WP (cf. ~20% from the EP). From December to May, the moisture transported to the TGRR is dominated by the westerlies, with relatively stable contribution from the two parts (89% from the WP and 11% from the EP). From May to September, as the Indian monsoon and the East Asia monsoon start, the moisture from the WP gradually decreases with the interaction of the two monsoon systems, and it

reaches annual minimum in September (61%). It is noteworthy that the moisture contribution of WP dominants the TGRR precipitation all year round.

We then estimate the moisture contributions of ocean (ocean sources), land (land sources, excluding the TGRR), and local recycling (TGRR sources) to the monthly precipitation in the TGRR (Fig. 5b). On average, 41% of the moisture is from

ocean sources, 56% from land sources, and only 3% from local recycling (recycling within the target region). Similar to the results for WP and EP, the contributions of ocean sources, land sources, and local recycling are stable from December to May (38%, 60%, and 2%, respectively), during which the land sources are predominant. However, during the summer monsoon season, oceans become the dominant sources with the moisture contribution reaching its maximum in June (54%). Meanwhile, the contribution of land sources drops to its minimum (43% in July). From July to November, the moisture

contribution of ocean (land) sources gradually decreases (increases) as the monsoon systems recede. As a result, the terrestrial contribution reaches its minimum (67%) in November (cf. oceanic contribution is 30%). Different from that of land and ocean sources, the change of local recycling is dominated by the variations in the water surface, soil moisture, and local circulations within the target region. In particular, the contribution of local recycling peaks in August (5%), even though the maximum monthly precipitation occurs in July. Similar phenomenon has been reported by *Fremme and*

*Sodemann* (2019) in a relatively greater region (the entire YRB), mainly resulting from an interplay among multiple factors such as weakened winds, high soil moisture, dense vegetation cover, and high evaporation. Overall, land sources dominate the moisture contributing to the TGRR precipitation, except during the summer monsoon season.

**3.2 Temporal trends of moisture sources of precipitation in the TGRR**

In this section, we investigate the impacts of spatiotemporal changes in moisture source contributions on the decreasing

precipitation in the TGRR (see Fig. 2). Figure 6 shows the trends in the contribution of the moisture transported from




different regions to the annual precipitation in the TGRR estimated with the simple linear regression model (stippling represents regions with statistically significant trends; $p<0.05$). It is clear that the WP is dominated by decreasing (negative) trends of moisture contributions. The highest decreasing trends are observed in the southeastern tip of the TP (>1.0 mm/decade, $p<0.05$). In addition, the negative trends of moisture contribution are statistically significant ($p<0.05$) along the

southern slope of the TP, over the northwestern Bay of Bengal as well as (almost) the entire Arabian Sea. In general, source regions with decreased trends of moisture contribution are mainly dominated by the Indian monsoon and the southern branch of middle-latitude westerlies. In contrast, regions with increasing (positive) trends of moisture contribution are relatively small in size, primarily scattered in the regions south and east of the TGRR. A strip with the greatest increasing trends (>0.4 mm/decade) appears between 108°E and 114°E. Nevertheless, the vast majority of these increasing trends are statistically

insignificant ($p\geq0.05$). It is interesting that the major source regions with positive and negative trends are separated by a north–south line, which is roughly in line with the 108°E used for WP and EP (see Fig. 3). This distinct division may reflect the interaction of different circulation systems that transport moisture to the TGRR in different seasons.

We then analyze the seasonal variations in the changing contribution of moisture from different source regions, with the

results summarized in Fig. 7. In spring, source regions with significant decreasing trends of moisture contribution form a southwest–northeast band extending from the southeastern TP to the northern Bay of Bengal. The spatial distribution of the trends in moisture contribution during summer largely resembles that of the annual precipitation (Fig. 6). With the onset of the monsoon season, the source regions with negative trends shift southwards (cf. Fig. 4b), which cover nearly the entire WP land, the northwestern Bay of Bengal and part of the Arabian Sea. Meanwhile, a strip with insignificant increasing trends of

moisture contribution is observed to the east of 108°E. Among all source regions in autumn, the regions with significant negative trends are distributed around the southeastern tip of the TP, while those with increasing trends are seen in regions south and southeast of the TGRR. During winter, only source regions southwest of the TGRR show significant decreasing trends of moisture contribution, which is much weaker than in other three seasons. The seasonal analyses reveal that although with seasonal variations, the decreased precipitation in the TGRR during 1979–2015 is mainly attributable to the

decreased contribution of moisture from regions southwest of the target region, especially around the southeastern tip of the TP.

We further define two key source regions based on the trends shown in Fig. 6: the southwestern source (SWS; 23°N to 29°N and 94°E to 104°E) and southern source (SS; 22°N to 29°N and 108°E to 112°E) regions. The interannual variations of their

contributions to the change in TGRR annual precipitation are shown in Fig. 8. For comparison purposes, the corresponding time series of local recycling (%) and TGRR precipitation (mm) are also included. Similar to the results in Figs. 6 and 7, the decreasing trend of moisture contribution from the SWS region (–1.1 mm/decade; $p<0.05$) is much stronger than the increasing trend for the SS region (0.3 mm/decade). The contribution of local recycling also shows an increasing trend (0.1%/decade), indicating that moisture contribution from the internal TGRR may increase faster than that from the external





during the period. This is also in line with the increasing trend of reference evapotranspiration during 1982–2013 observed in
*Lv et al.* (2016). The values of Pearson correlation coefficient between annual TGRR precipitation and the moisture from the
SWS region, the moisture from the SS region, and the contribution of local recycling are 0.68 ($p<0.05$), 0.28, and –0.61
($p<0.05$), respectively. This suggests that the fluctuation of annual precipitation in the TGRR can be largely explained by the
interannual variation of moisture contribution from the SWS region. Meanwhile, the high (low) annual precipitation in the
TGRR is often accompanied by a low (high) contribution from the local recycling.

### 3.3 Contributions of moisture sources during wet and dry years

The anomalies in moisture contributions during some years may suggest the changes in the moisture transport during
extreme events, such as floods and droughts. In this section, we investigate the difference in the contributions of moisture
sources during wet and dry years. We select three wet years (1989, 1998, and 2007) and three dry years (1997, 2001, and
2006) as the representative years based on the annual precipitation time series from CMA observations (Fig. 2b). The
seasonal variations of TGRR precipitation (mm) in the representative wet and dry years are shown in Fig. 9a. It is clear that
the rainfall difference between these two types mainly in April–September, with the largest difference in summer.

The differences in moisture contribution and vertical integrated moisture fluxes between the representative wet and dry years
(wet – dry) are shown in Fig. 9b. On average, the extra moisture during wet years (positive signals in Fig. 9b) are mainly
from the regions southwest of the TGGR, although some weak negative signals are seen in part of the adjacent regions
southeast of the target region. Note that this dipole pattern is also consistent with the trends of moisture contribution in Fig. 6.
On the other hand, intensified moisture transports from both the northern Indian Ocean and the northwest Pacific Ocean to
the eastern Bay of Bengal (along ~10°N) are observed during wet years. The oceanic moisture converges over the Bay of
Bengal and is then delivered to the target region by the southwesterly monsoon winds, forming an anticyclone pattern with
its center around the southwestern China. The patterns of the differences in summer moisture contribution between wet and
dry years (Fig. 9c) are very similar to their annual counterparts (Fig. 9b). However, the change in summer moisture fluxes
during wet and dry years are different from that of the annual mean. During summer, we only observe intensified moisture
transport from the northwest Pacific Ocean to the eastern Bay of Bengal in wet years, while the transport from the northern
Indian Ocean shows marginal changes. The intensified low-latitude westward moisture fluxes during summer in wet years
are in consistent with a previous numerical simulation focusing on flooding years over the entire YRB (*Xu et al.*, 2008).
Generally, compared to dry years, more moisture from regions southwest of the TGRR contributes to the higher TGRR
precipitation in wet years, with significantly intensified moisture transport from the low-latitude oceans.



## 4 Discussion

### 4.1 Uncertainties of moisture contribution with different reanalysis datasets

It is noteworthy that the accuracy of WAM-2layers can be affected by the resolution and reliability of the forcing data, as WAM-2layers is essentially a mass balance model that treats the globe as grids of columns (*Van der Ent et al.*, 2010; *Li et al.*, 2019a). The long-term trend in WAM-2layers simulations is also inevitably inherited from the forcing data. Existing studies on the moisture transport over the YRB are often based on single reanalysis product (*Xu et al.*, 2004; *Wei et al.*, 2012; *Chen et al.*, 2013; *Wang et al.*, 2018; *Fremme and Sodemann*, 2019). However, uncertainties in these datasets have been observed at both regional and global scales when estimating climate and hydrological changes (*Trenberth et al.*, 2011; *Lorenz et al.*, 2014; *Bosilovich et al.*, 2017). To better understand the uncertainties induced by reanalysis datasets (as forcing), similar moisture tracking simulations have been performed based on JRA-55 and MERRA-2 (see Section 2.2).

The spatial contributions of moisture to the annual precipitation in TGRR when using JRA-55 (1979–2015) and MERRA-2 (1980–2015) as the forcing datasets are shown in Fig. 10a and b. The moisture from the regions southwest of the TGRR estimated using MERRA-2 is considerably higher than those using ERA-Interim and JRA-55, which may result from the relatively higher difference between evaporation and precipitation over these regions in MERRA-2 (*Bosilovich et al.*, 2017). Nevertheless, simulations based on three forcing datasets (ERA-Interim, JRA-55, and MERRA-2) exhibit highly consistent spatial distributions of moisture source regions and transport pathways, suggesting the robustness of the major findings in Section 3.

The trends in the contribution when using JRA-55 and MERRA-2 are shown in Fig. 10c and d. Compared to the results shown in Fig. 6, the spatial distribution of the moisture contribution using JRA-55 shows general agreement with that using ERA-Interim, although with relatively weaker decreasing trends of moisture contribution over the SWS regions. In addition, the significant negative trends over the Arabian Sea observed in Fig. 6 is not found in Fig. 10c. In contrast, the trends estimated using MERRA-2 (Fig. 10d) show relatively large discrepancies when compared to that using ERA-Interim, especially the significant increasing trends over the southwestern edge of the TP and the regions north of the TGRR. The discrepancies in contributions and trends are due primarily to the inherent differences in the assimilation system, input data, and model physics of different reanalysis datasets. Nevertheless, clear dipole patterns of positive and negative signals in SWS and SS regions emerge in all three sets of simulations (Figs. 6, 10c, and 10d). Overall, the three forcing datasets show generally good agreement when identifying the spatial characteristics of major source regions and the variations in the contributions to the TGRR precipitation. Such consistency further demonstrates the capacity and accuracy of ERA-Interim-based moisture tracking in detecting source regions and their contributions to the precipitation in the target region.



### 4.2 Potential impacts of the TGR on the moisture from local recycling

The operation of the TGR since its first impoundment in 2003 results in a dramatic expansion of the water surface in the TGRR. Although most existing work has suggested that the long-term impact of TGR on local climate (e.g., precipitation and extreme droughts and floods) is insignificant (*Xiao et al.*, 2010; *Xu et al.*, 2013; *Lü et al.*, 2018), some studies did find that the fast expansion of the water surface tends to decrease surface temperature, increase regional humidity, enhance the descending motion of air, and alter the regional atmospheric circulation under some special synoptic conditions (*Miller et al.*, 2005; *Li et al.*, 2019b; *Zeng et al.*, 2019). On the other hand, local recycling is a diagnostic measure that depicts the surface hydrology and regional climate in a target region. For example, it plays an important role in sustaining precipitation in the later part of the monsoon season in the YRB (*Fremme and Sodemann*, 2019). Here we focus on the change of local recycling before and after the first impoundment of the TGR (in 2003) and investigate the potential impacts of this water conservation project.

Figure 11 shows the temporal change of moisture from the TGRR (local recycling; %) and its contribution (local contribution; mm) to the annual precipitation in the TGRR before and after 2003. Both the moisture from local recycling and its contribution show significant increasing trends before 2003 (0.2%/decade and 4.2 mm/decade, respectively; $p < 0.05$). This is followed by a sudden drop in both variables right after the first impoundment of the TGR in 2003. The weakened local recycling may result from the enhanced sinking air and moisture divergence in the lower atmosphere with surface cooling (*Miller et al.*, 2005). In contrast, no significant trends are found in these two variables during 2003–2015, partly because of the limited temporal coverage. Nevertheless, it is still not clear whether the abrupt change in the local recycling is dominated by the oscillation of the large-scale circulations or the impoundment of the TGR, and the mechanisms of this sudden drop require further investigation.

### 5 Conclusions

Moisture transport plays an essential role in the local precipitation seasonality and its long-term change. In this study, we investigated the long-term (1979–2015) moisture sources that contribute to the annual and seasonal precipitation change over the TGRR based on numerical simulations. We examined in detail the variations of major source regions and the interactions between different circulation systems. The contribution of moisture to TGRR precipitation in representative wet and dry years was further evaluated. In addition, we assessed the possible uncertainties induced by different forcing data. Potential impacts of the TGR were also studies based on the time series of local recycling. The main conclusions are:

(1) The source regions southwest of the TGRR (including the southeastern tip of the TP) play a dominate role in providing moisture to the annual and seasonal precipitation in the TGRR. In cold seasons (spring and winter), source regions mainly stretch westwards along the southern branch of the westerlies. Source regions are significantly greater in summer than in





other three seasons, with the Indian monsoon bringing moisture even from the Arabian Sea. In autumn, most source regions are mainly distributed over the land due to the gradual withdrawal of the monsoon systems.

(2) On average, the WP (west of 108°E) and EP (east of 108°E) of the domain contributes 80% and 20% of the TGRR precipitation, respectively. The contributions of ocean sources, land sources (excludes the TGRR), and local recycling are 41%, 56%, and 3%, respectively.

(3) The decreased annual precipitation in the TGRR during 1979–2015 is mainly due to the decreased contribution of the
moisture from source regions southwest of the TGRR, especially around the southeastern tip of the TP. Similar signals are observed in all four seasons. A dipole pattern of the increasing and decreasing trends is found between the SS and SWS regions, with the trend of moisture contribution over the SWS regions being –1.1 mm/decade ($p<0.05$). Correlation analyses further suggest that the change of moisture from the SWS regions dominates the annual precipitation trend and its fluctuation over the TGRR.


(4) The extra moisture during wet years is primarily from the regions southwest of the TGRR. Compared to dry years, intensified moisture transport is observed from the low-latitude oceans during wet years; the moisture converges over the Bay of Bengal and is eventually delivered to the target region by the southwesterly monsoon winds.

The findings of this study reveal the major source regions of the TGRR precipitation and their spatial and temporal dynamics, providing solid evidence for the response of regional hydrological cycle to climate change and the construction of the TGD. This study also has much broader implications beyond the TGRR. On the one hand, most studies on regional precipitation change still heavily rely on station measurements or satellite datasets and are therefore relatively restricted to the quantification of local impacts. The use of the atmospheric moisture tracking model here exemplifies the contributions of
nonlocal moisture sources to the regional precipitation change, highlighting the need for moisture tracking in quantifying hydrological changes in other regions of the world. On the other hand, atmospheric moisture tracking can help clarify the debate on the potential hydrological and environmental impacts of dam construction and large reservoirs (like the TGR). Furthermore, the changes in source regions and their contributions in different seasons and years (dry and wet) suggest potential blind spots in some existing regional water resource management and disaster prevention practices. For example,
the change of moisture source induced by land-use policies in nearby states, provinces, or countries may directly influence the precipitation over the target region. It is necessary to take into account the impact of long-distance moisture transport in regional water resource planning and decision-making processes.

**Data availability**

The global atmospheric reanalysis product, ERA-Interim, can be downloaded from the official website of the European
Centre for Medium-Range Weather Forecasts (ECMWF; https://www.ecmwf.int/en/forecasts/datasets/reanalysis-datasets/era-interim). The MERRA-2 dataset is available from https://gmao.gsfc.nasa.gov/reanalysis/MERRA-2/, which is managed by the Goddard Earth Sciences Data and Information Services Center (GES DISC), National Aeronautics and Space Administration (NASA). The JRA-55 product was developed by the Japan Meteorological Agency and can be downloaded from https://jra.kishou.go.jp/. The interpolated precipitation dataset from China Meteorological Administration
was retrieved from the official website of the China Meteorological Data Service Center (CMDC; http://data.cma.cn). The TMPA dataset is also managed by the NASA's GES DISC, available from https://disc.gsfc.nasa.gov.

**Author contribution**

YL, CW, HP, SX, and DY conceptualized the study. YL carried out numerical simulations, conducted formal analysis, and prepared figures, and wrote the initial draft. CW contributed to the visualization of results. All authors contributed to the
review and editing of the manuscript.

**Competing interests**

The authors declare that they have no conflict of interest.

**Acknowledgements**

This work was supported by the National Key Research and Development Project (Grant No. 2016YFA0601503) and the
National Science Fund for Distinguished Young Scholars (Grant No. 51725905).

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



**Figures**

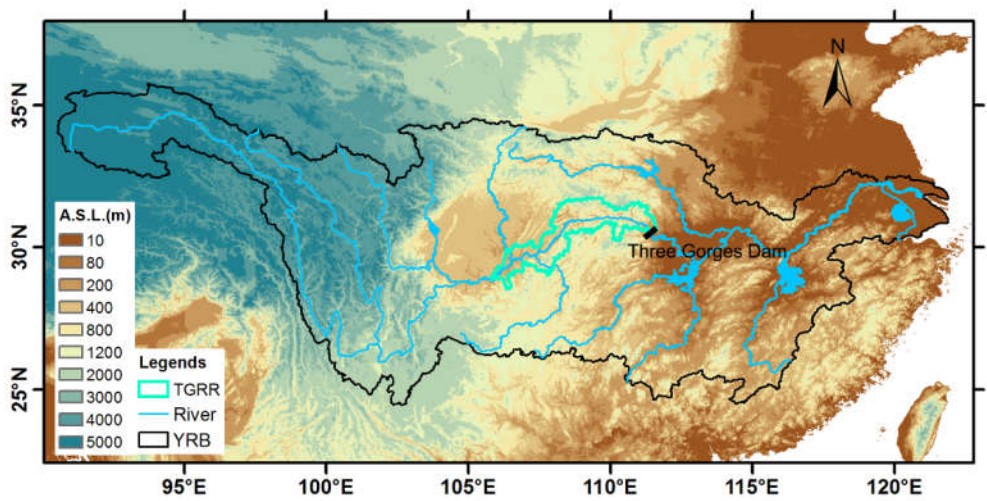

**Figure 1:** Topography of the Yangtze River Basin (YRB, black line), with the cyan boundary denoting the Three Gorges Reservoir Region
(TGRR).

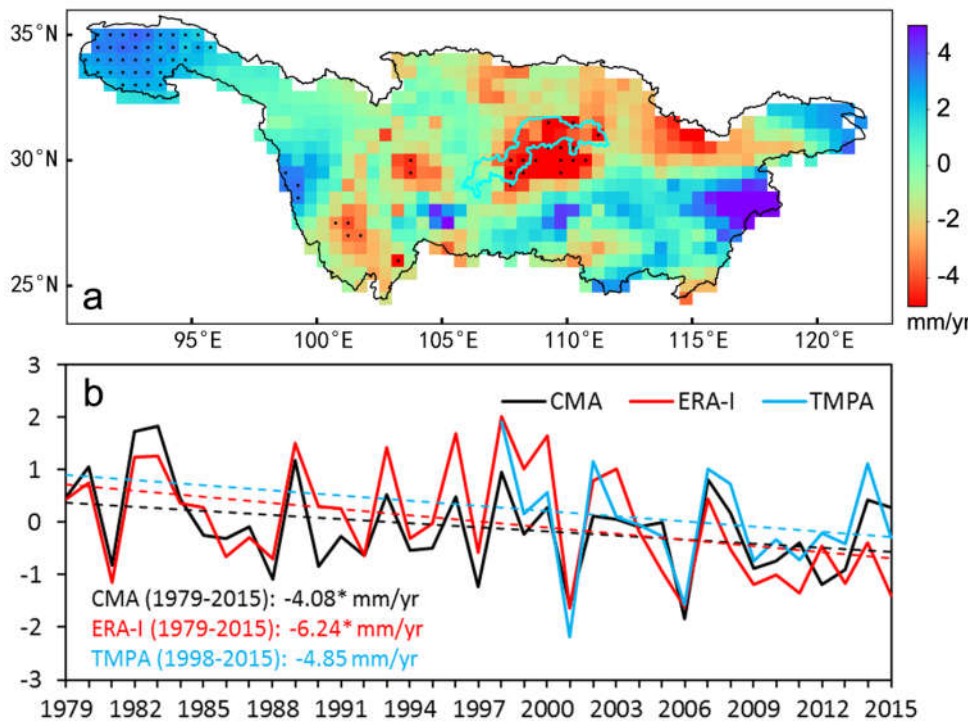

**Figure 2:** Spatial and temporal change of annual precipitation in the Yangtze River Basin (YRB) and the Three Gorges Reservoir Region
(TGRR). (a) Spatial distribution of linear trends in annual precipitation over the YRB during 1979–2015 based on CMA observations. (b)





Time series of the standardized annual precipitation (Z-scores) over the TGRR based on CMA (1979–2015), ERA-Interim (1979–2015), and TMPA (1998–2015). Stippling in (a) indicates regions with statistically significant trends ($p < 0.05$). Dashed lines in (b) are linear regression fits to the data. The corresponding slopes are also shown, with the asterisks for statistically significant trends ($p < 0.05$).

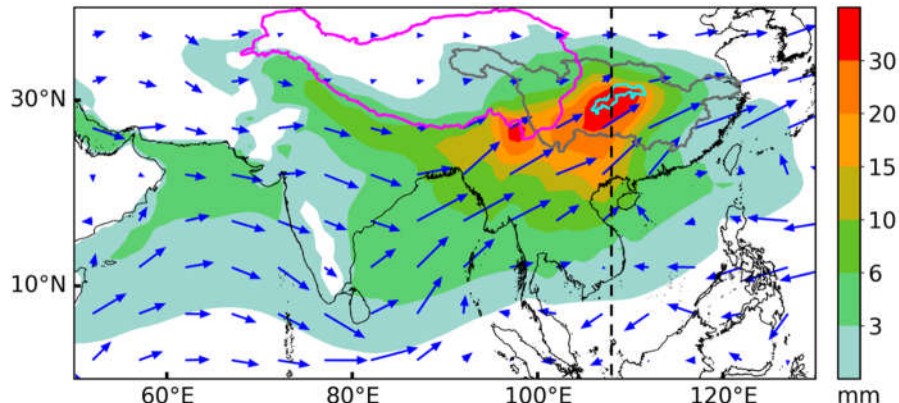

**Figure 3:** Contribution of the moisture (mm) from different source regions to the annual precipitation in the Three Gorges Reservoir Region (TGRR) averaged over 1979–2015, overlaid with the patterns of vertical integrated moisture fluxes (blue arrows). The purple line denotes the boundary of the Tibetan Plateau (TP), and the gray line denotes the boundary of the Yangtze River Basin (YRB). The dotted black line shows 108°E that divides the domain into the western part (WP) and the eastern part (EP).

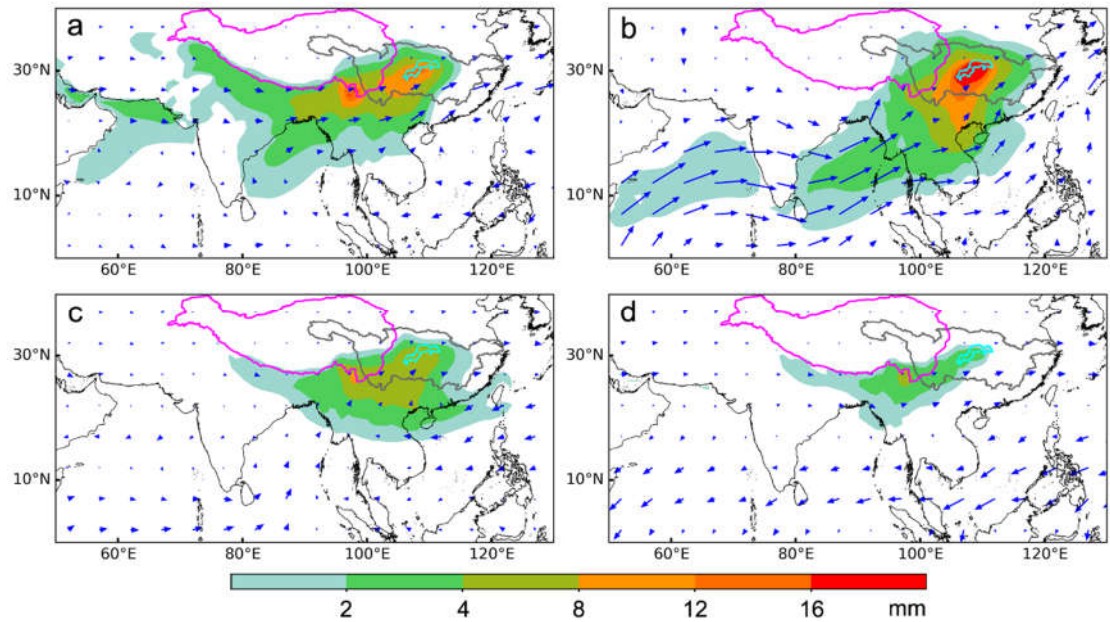


**Figure 4:** Same as Fig. 3 but for seasonal precipitation: (a) spring, (b) summer, (c) autumn, and (d) winter averaged over 1979–2015.

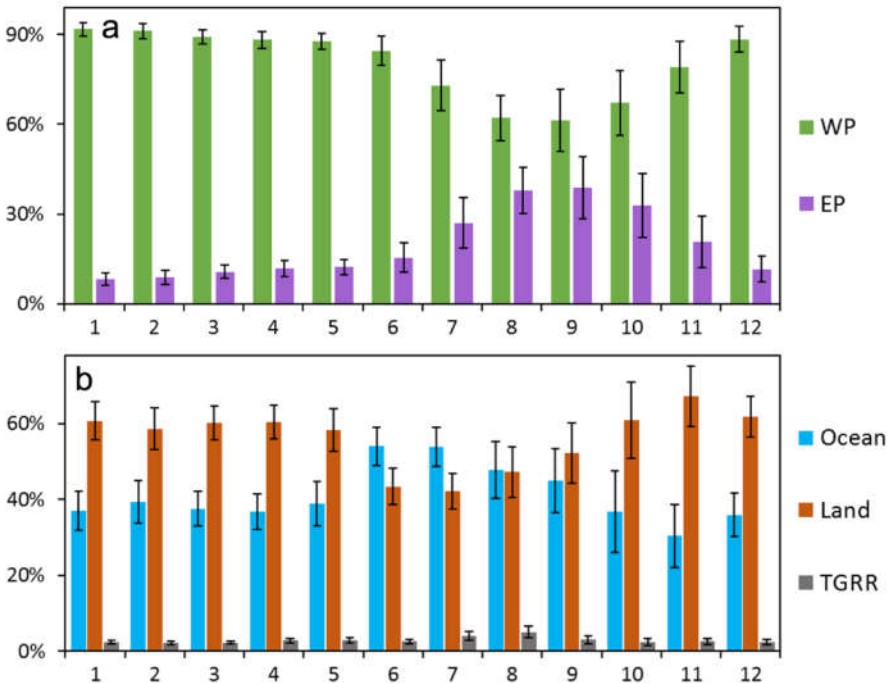

**Figure 5:** Contribution of different sources (%) to the monthly precipitation in the Three Gorges Reservoir Region (TGRR) averaged over 1979–2015: (a) western part (WP) and the eastern part (EP) of the domain, and (b) ocean, land, and TGRR (local recycling). Error-bars represent one standard deviation of the interannual variations.

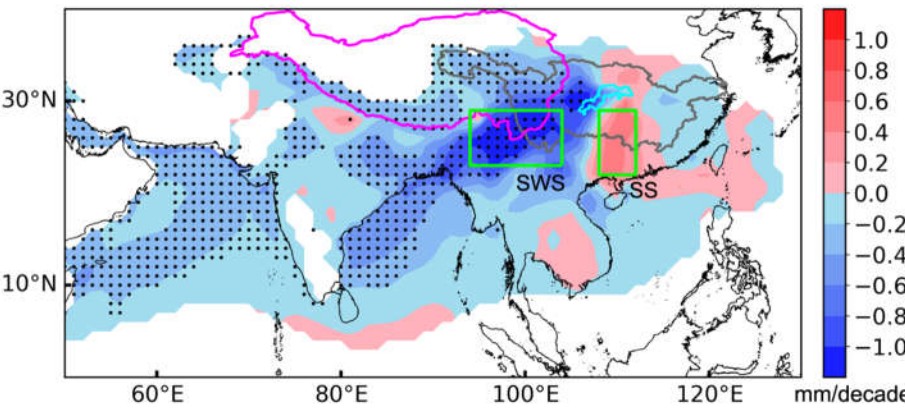

**Figure 6:** Trends in the contribution of the moisture (mm/decade) from different source regions to the annual precipitation in the Three Gorges Reservoir Region (TGRR) averaged over 1979–2015. Stippling indicates regions with statistically significant trends ($p < 0.05$). Green boxes represent the southwestern source (SWS) and southern source (SS) regions.

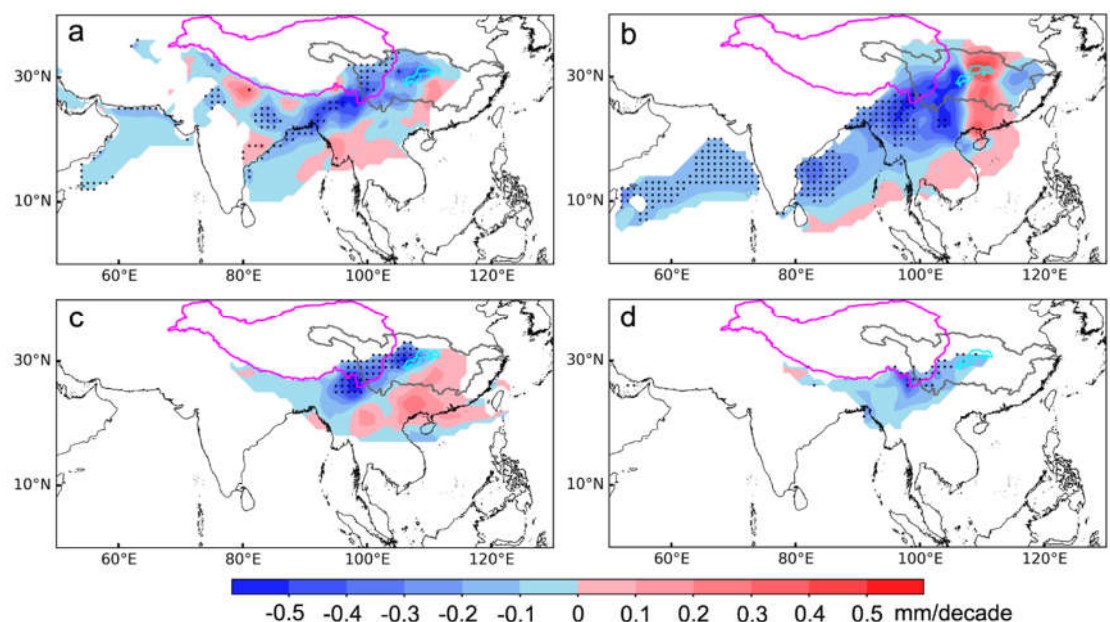

**Figure 7:** Same as Fig. 6 but for seasonal precipitation: (a) spring, (b) summer, (c) autumn, and (d) winter averaged over 1979–2015.


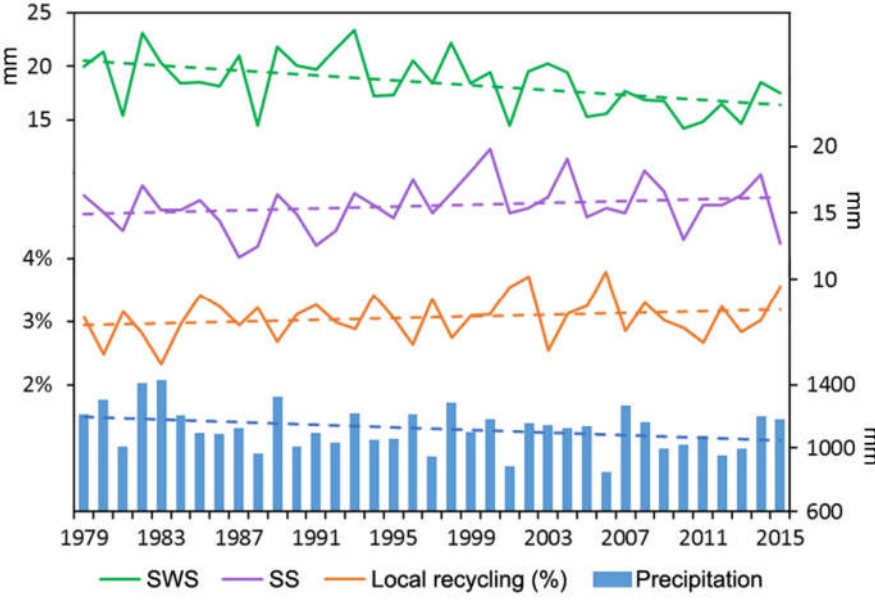

**Figure 8:** Temporal change of moisture (mm) from the southwestern source (SWS) and southern source (SS) regions (see Fig. 6) and the local recycling (%) to the annual precipitation (mm) in the Three Gorges Reservoir Region (TGRR). Dashed lines are linear regression fits to the data.


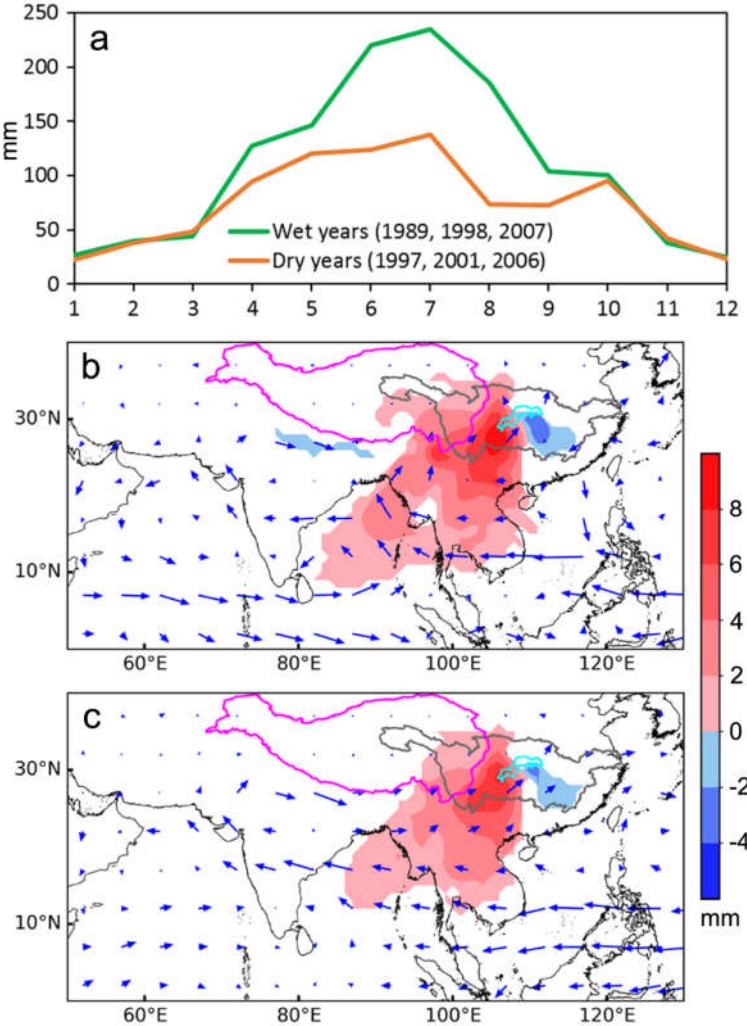

**Figure 9:** Precipitation and contribution of moisture in wet and dry years: (a) seasonal variation of the Three Gorges Reservoir Region (TGRR) precipitation (mm) in wet (1989, 1998, and 2007) and dry (1997, 2001, and 2006) years, and the difference in moisture (mm) contributing to (b) annual and (c) summer precipitation in the TGRR between wet and dry years (wet – dry), overlaid with the difference in vertical integrated moisture fluxes (blue arrows; wet – dry).

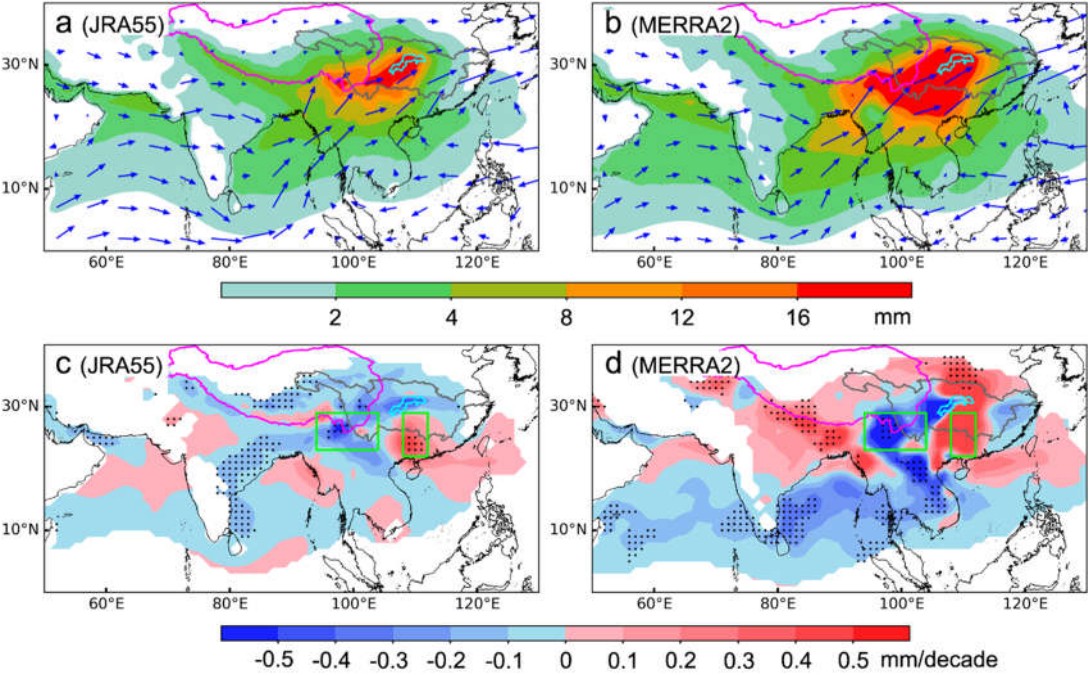

**Figure 10:** (a) and (b) Contributions of the moisture (mm) from different source regions to the annual precipitation in TGRR, and (c) and (d) their trends (mm/decade) when using two reanalysis datasets as forcing: (a) and (c) JRA-55 (1979–2015); (b) and (d) MERRA-2 (1980–2015). Blue arrows in (a) and (b) are vertical integrated moisture fluxes. Stippling in (c) and (d) indicates regions with statistically significant trends ($p < 0.05$).

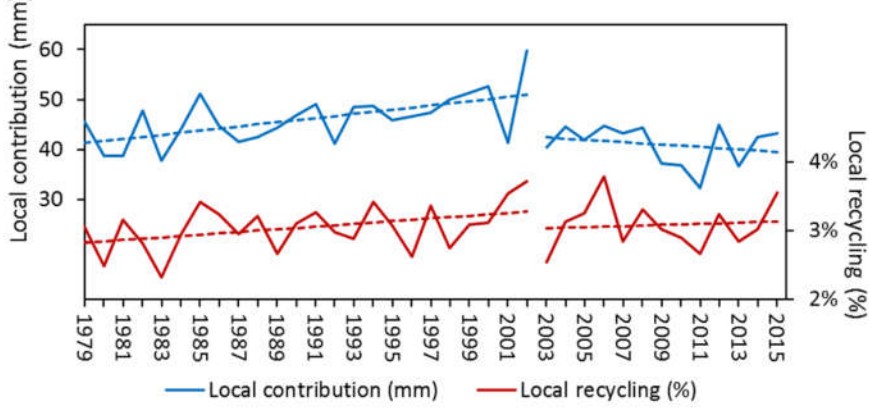

**Figure 11:** Temporal change of moisture from the TGRR (local recycling; %) and its contribution (local contribution; mm) to the annual precipitation in the Three Gorges Reservoir Region (TGRR) before and after the first impoundment of the Three Gorges Reservoir (TGR) in 2003. Dashed lines are linear regression fits to the data.