# Peer review of "Contribution of moisture sources to precipitation changes in the Three Gorges Reservoir Region"

_Hydrology and Earth System Sciences, 2020_

## Referee Comment (RC1) · Anonymous Referee #1 · 2 Feb 2021

This paper has addressed an important scientific question regarding the moisture sources contribution to the change in precipitation over the Three Gorges Reservoir Region. Title and abstract reflect the content of the paper clearly. This paper has outlined the details about the data, model and methodology clearly. My main concern is about the conclusions derived from the results. • In general, this geographic region (TGRR) is influenced by the south-west monsoon (mainly in summer) which brings moisture from the Arabian Sea and Bay of Bengal, as is suggested by figures 3 and 4. Results also suggest (as is expected) the influence of westerlies system on the moisture source region. Then, the authors claim that the decreased precipitation over the TGRR during 1979-2015 is mainly due to the decreased moisture contribution from

the source regions which span over the southwest of the TGRR, especially around the southeastern tip of the Tibetan Plateau. In order to establish the above-mentioned causation, authors presented trend analyses (figures 6,7 ) and the time series (figure 8), which suggest the association between the SWS region and the TGRR. No physical mechanism is provided to explain how the decreasing moisture contribution from the SWS region (including the southeastern tip of the Tibetan Plateau) leads to the decreasing precipitation over the TGRR. In other words, it is not clear what is the pathway of this causation. Furthermore, is this the case only for summer? • How does this conclusion hold true during the winter and spring when the westerlies influence is strong? This part is not clear as well . • Authors have selected two boxes to identify two source regions, i.e. SWS and SS. The result of figure 8 depends a lot on how the box is defined. • What do we know about the general variability and trend in moisture over the SWS region? • Some form of visualization/analysis on a large-scale map (focusing TGRR) would help the readers in later section. In summary, the use of moisture tracking model in identifying the dominant source regions of moisture is useful. Hence, the first part of the conclusions is convincing and clear. But another key point of the conclusions associated with the role of specific geographic region as a source of moisture in increasing/decreasing the precipitation over the TGRR is not clear.

---

## Referee Comment (RC2) · Anonymous Referee #2 · 26 Feb 2021

This paper studies the contribution of moisture sources to precipitation in the TGR region from 1979 through 2015. It does so by numerically tracking the moisture sources of annual and seasonal precipitation, by identifying the main sources that determine the interannual variability of precipitation and by analyzing the differences in moisture sources and their transport during extreme years over the region. The paper illustrates the trends in annual precipitation and the annual and seasonal variability very well over TGRR. To further look into the sources of moisture, the study divides the region into western and eastern parts and uses the moisture tracking method over the two domains to quantify the contribution from different regions. Contribution to the WP is thought to mainly come from the Indian monsoon system where as the EP comes

from the EA monsoon system. In this case how the region is divided into the two domains could influence the results, some more analysis of the domain extent could establish this relation further. Similarly, contribution from ocean, land and local recycling are also analyzed, showing land sources being dominant, these results are more convincing. The spatiotemporal trends of moisture sources on precipitation with decreasing trends of moisture contribution mainly come from Indian monsoon and with a marginally increasing trend of moisture contribution from local recycling - all good reasoning, however, correlating the annual TGRR precipitation to the different regional moisture sources is not enough to say that the variability in these moisture sources cause the changes in TGRR precipitation. The analysis of extremes has just 3 sample years each for wet and dry conditions, can the sample size be increased? The paper used the Water Accounting Model-2layers to simulate the moisture transport and to quantify and pinpoint the sources of moisture well, although the method has limitations it has been used quite effectively in this study.

---

## Author Comment (AC1) · 4 Mar 2021

**Response to Anonymous Referee #1**

**Comment:** *This paper has addressed an important scientific question regarding the moisture sources contribution to the change in precipitation over the Three Gorges Reservoir Region. Title and abstract reflect the content of the paper clearly. This paper has outlined the details about the data, model and methodology clearly. My main concern is about the conclusions derived from the results. In general, this geographic region (TGRR) is influenced by the south-west monsoon (mainly in summer) which brings moisture from the Arabian Sea and Bay of Bengal, as is suggested by figures 3 and 4. Results also suggest (as is expected) the influence of westerlies system on the moisture source region. Then, the authors claim that the decreased precipitation over the TGRR during 1979-2015 is mainly due to the decreased moisture contribution from the source regions which span over the southwest of the TGRR, especially around the southeastern tip of the Tibetan Plateau.*

**Response:** Thank you for your constructive comments and suggestions. Please see our responses below.

**Comment:** *In order to establish the above-mentioned causation, authors presented trend analyses (figures 6, 7) and the time series (figure 8), which suggest the association between the SWS region and the TGRR. No physical mechanism is provided to explain how the decreasing moisture contribution from the SWS region (including the southeastern tip of the Tibetan Plateau) leads to the decreasing precipitation over the TGRR. In other words, it is not clear what is the pathway of this causation.*

**Response:** Thank you for your comment. First of all, we would like to point out that the WAM-2layers model we used in this study is a moisture tracking model that numerically determines the moisture origin based on mass balance (see, e.g., van der Ent et al., 2013). We drew this conclusion mainly because, among all source regions with decreasing trends in annual and seasonal moisture contribution (Figs. 6 and 7, cf. Figs. 3 and 4), most are concentrated within SWS. In contrast, SS mainly experienced increasing trends in moisture contribution (although statistically insignificant). We then used the correlation results in Fig. 8 to show the relationship between variations of annual precipitation and moisture contributions.

Per your comment, we will further elaborate on the underlying mechanisms of this causal relationship in the revision. We will discuss this based on evaporation, vertical integrated moisture flux in the zonal direction, and vertical integrated moisture flux in the meridional direction based on reanalysis data. Figure R1 below shows the changes (trends) of these three fields in 1979–2015 and the two major pathways of moisture transport toward the target region (arrows in yellow). These two major pathways (from northern India Ocean to TGRR and from northwest Pacific Ocean to TGRR) are primarily based on Fig. 3, which have been identified in previous studies in Yangtze River basin (e.g., Xu et al., 2014). Note that the two key source regions, SWS and SS, are located along these two pathways and are critical to the moisture transport. Figure R2 shows 800-hPa, 500-hPa, and 300-hPa wind fields and their trends over time.

[Figure]

**Figure R1**: Trends of (a) evaporation, (b) vertical integrated moisture flux in the meridional direction (positive northward), and (c) vertical integrated moisture flux in the zonal direction (positive eastward) based on ERA-Interim during 1979–2015. Arrows in yellows show two major pathways of moisture transport toward the target region.

It is clear that the vast majority of all possible source regions experienced increased evaporation during the study period, despite a small portion of the SWS region with statistically insignificant decreased evaporation (Fig. R1a). Therefore, the enhanced evaporation increase over the SWS is unlikely the major cause of decreased precipitation in the TGRR. We then turn to vertical integrated moisture fluxes. As shown in Figs. 3 and R2, the two major pathways of moisture transport are controlled by winds at different pressure heights. The southwest pathway (from northern Indian Ocean) is mainly controlled by winds at relatively lower levels, while the southeast pathways (from northwest Pacific Ocean) is mainly controlled by winds at relatively higher levels. For the southwest pathway (from northern Indian Ocean), northward and eastward vertical integrated moisture fluxes in general enhanced along the pathway before reaching the SWS region (Fig. R1b and c). However, the further transport of moisture toward the TGRR is largely dampened by the decreased northward and eastward moisture flux over the eastern part of the SWS region, which contributes to the decreased precipitation in the TGRR. For the southeast pathway (from northwest Pacific Ocean), largely decreased eastward moisture flux over the northwest Pacific Ocean and South China Sea indicates an increased westward moisture contribution to the

target region. But this enhancement is partly offset by the decreased northward moisture flux along the pathway (especially over the SS region), which results in statistically insignificant trends as observed in Figs. 6 and 7.

[Figure]

**Figure R2**: Trends of 800-hpa (top), 500-hpa (middle), and 300-hpa (bottom) wind in the (a, b, and c) meridional direction (positive northward) and (d, e, and f) zonal direction (positive eastward) based on ERA-Interim during 1979–2015, overlaid with wind vectors (yellow arrows).

We will add the above analysis and Fig. R1 in the revision. Nevertheless, our understanding can be partly limited by the moisture tracking model as well as our selected reanalysis dataset used in this study. We will rely on more sophisticated models to investigate the dynamics of specific systems (e.g., monsoon system) for the same period in future study.

**Comment:** *Furthermore, is this the case only for summer? How does this conclusion hold true during the winter and spring when the westerlies influence is strong? This part is not clear as well.*

**Response:** Thank you for the question. We have analyzed seasonal trends. The annual and seasonal trends of moisture contributions and TGRR precipitation are summarized in Table R1. As mentioned in our original submission, the decreasing trend in annual precipitation is consistent with decreased moisture contribution from the SWS region. Similar consistency is observed also for summer, autumn, and winter, although statistically significant trends ($p < 0.05$) only occur in

summer. The weak relationship between precipitation and SWS moisture contribution in spring, autumn, and winter are likely due to the absence (or marginal influence) of the Indian monsoon, during which the westerlies influence is stronger. We will add these seasonal analyses and Table R1 in the revision.

**Table R1:** Trends of precipitation over the TGRR and moisture contribution from SWS, SS, and local recycling, on annual and seasonal scales during 1979–2015. '*' represents statistically significant trends ($p < 0.05$).

| | Precipitation (mm/decade) | SWS (mm/decade) | SS (mm/decade) | Local recycling (%/decade) |
|---|---|---|---|---|
| Annual | –40.81* | –9.16* | 1.45 | 0.04 |
| Spring | 5.60 | –2.22 | 0.21 | 0.02 |
| Summer | –35.67* | –3.41* | 0.95 | 0.14 |
| Autumn | –7.90 | –2.26 | 0.46 | –0.06 |
| Winter | –2.83 | –1.27 | –0.17 | 0.08 |

**Comment:** *Authors have selected two boxes to identify two source regions, i.e., SWS and SS. The result of figure 8 depends a lot on how the box is defined. What do we know about the general variability and trend in moisture over the SWS region?*

**Response:** This is a good question. The two key regions are defined because (1) they experienced most significant changes in moisture contribution as shown in Fig. 6, and (2) they are located along the two major pathways as shown in Fig. R1 (will be added in the revision) and Fig. 3. To test the potential uncertainties induced by the selection of the bounding boxes, we further quantified the trends of moisture contribution from WP and EP (defined in Section 3.1) and their relationships with precipitation in TGRR. Results are shown in Fig. R3 and Table R2.

[Figure]

**Figure R3:** Temporal change of moisture (mm) from the WP, EP, SWS, SS and local recycling (%) to the annual precipitation (mm) in the TGRR. Dashed lines are linear regression fits to the data.

**Table R2:** Trends of moisture contributions and correlation coefficients between annual TGRR precipitation and moisture contributions in 1979–2015. '*' represents statistically significant (p < 0.05).

| | Precipitation (mm/decade) | WP (mm/decade) | EP (mm/decade) | SWS (mm/decade) | SS (mm/decade) | Local recycling (%/decade) |
|---|---|---|---|---|---|---|
| Trend | −40.81* | −60.33* | −2.37 | −9.16* | 1.45 | 0.04 |
| Correlation coefficient | — | 0.69* | 0.34* | 0.68* | 0.28 | −0.64* |

It is clear that the increasing moisture contribution over the WP is consistent with that over SWS, while contributions from both EP and SS regions show very marginal and statistically insignificant changes. Our conclusions are therefore not affected by the boxes we selected. We will add these analyses, Fig. R3, and Table R2 in the revision. In particular, Fig. 8 in the main text will be replaced by Fig. R3.

Please refer to our previous response (Fig. R1) for the changes of moisture flux and evaporation over the SWS region.

**Comment:** *Some form of visualization/analysis on a large-scale map (focusing TGRR) would help the readers in later section.*

**Response:** Thank you for the suggestion. We have tried large-scale maps focusing on TGRR, but due to the coarse resolution of the moisture tracking model (as determined by numerical stability and computational cost, see Section 2.1), large-scale maps do not show any additional details. We will rely on other moisture tracking models (e.g., Lagrangian ones) to refine our simulations in the future.

**Comment:** *In summary, the use of moisture tracking model in identifying the dominant source regions of moisture is useful. Hence, the first part of the conclusions is convincing and clear. But another key point of the conclusions associated with the role of specific geographic region as a source of moisture in increasing/decreasing the precipitation over the TGRR is not clear.*

**Response:** Thank you for your valuable feedback. We will discuss the physical mechanisms of the role that SWS region plays in TGRR precipitation change in the revision (please refer to our responses above).

**References**

van der Ent, R. J., Tuinenburg, O. A., Knoche, H. R., Kunstmann, H., & Savenije, H. H. G. (2013). Should we use a simple or complex model for moisture recycling and atmospheric moisture tracking?. Hydrology and Earth System Sciences, 17(12), 4869-4884.

Xu, X., Chen, L., Wang, X., Miao, Q., & Tao, S. (2004). Moisture transport source/sink structure of the Meiyu rain belt along the Yangtze River valley. Chinese Science Bulletin, 49(2), 181-188.

---

## Author Comment (AC2) · 4 Mar 2021

**Response to Anonymous Referee #2**

**Comment:** *This paper studies the contribution of moisture sources to precipitation in the TGR region from 1979 through 2015. It does so by numerically tracking the moisture sources of annual and seasonal precipitation, by identifying the main sources that determine the interannual variability of precipitation and by analyzing the differences in moisture sources and their transport during extreme years over the region. The paper illustrates the trends in annual precipitation and the annual and seasonal variability very well over TGRR. To further look into the sources of moisture, the study divides the region into western and eastern parts and uses the moisture tracking method over the two domains to quantify the contribution from different regions. Contribution to the WP is thought to mainly come from the Indian monsoon system where as the EP comes from the EA monsoon system.*

**Response:** Thank you for your constructive comments and suggestions. Please see our responses below.

**Comment:** *In this case how the region is divided into the two domains could influence the results, some more analysis of the domain extent could establish this relation further.*

**Response:** The division of two (sub)domains was mainly informed by the two major moisture transport pathways observed in Fig. 3 and the dipole patterns observed in Figs. 6, 7, 9, 11. Note that the two major pathways (see Fig. R2), from northern India Ocean to TGRR and from northwest Pacific Ocean to TGRR, have also been identified in previous studies in Yangtze River basin (e.g., Xu et al., 2014). We have tested the selection of different subdomain sizes, but the results and conclusions are consistent. For example, Fig. R1 and Table R1 below show the temporal variations of moisture contributions from subdomains with different sizes and their relationship with TGRR precipitation. Note that the WP and EP are defined in Section 3.1.

[Figure]

**Figure R1:** Temporal change of moisture (mm) from the WP, EP, SWS, SS and local recycling (%) to the annual precipitation (mm) in the TGRR. Dashed lines are linear regression fits to the data.

**Table R1:** Trends of moisture contributions and correlation coefficients between annual TGRR precipitation and moisture contributions in 1979–2015. '*' represents statistically significant ($p < 0.05$).

|  | Precipitation (mm/decade) | WP (mm/decade) | EP (mm/decade) | SWS (mm/decade) | SS (mm/decade) | Local recycling (%/decade) |
|---|---|---|---|---|---|---|
| Trend | –40.81* | –60.33* | –2.37 | –9.16* | 1.45 | 0.04 |
| Correlation coefficient | — | 0.69* | 0.34* | 0.68* | 0.28 | –0.64* |

It is clear that the increasing moisture contribution over the WP is consistent with that over SWS, while contributions from both EP and SS regions show very marginal and statistically insignificant changes. Our conclusions are therefore not affected by the subdomain size we selected. We will add these analyses, Fig. R1, and Table R1 in the revision. In particular, Fig. 8 in the main text will be replaced by Fig. R1.

**Comment:** *Similarly, contribution from ocean, land and local recycling are also analyzed, showing land sources being dominant, these results are more convincing. The spatiotemporal*

*trends of moisture sources on precipitation with decreasing trends of moisture contribution mainly come from Indian monsoon and with a marginally increasing trend of moisture contribution from local recycling - all good reasoning, however, correlating the annual TGRR precipitation to the different regional moisture sources is not enough to say that the variability in these moisture sources cause the changes in TGRR precipitation.*

**Response:** Thank you for your comment. First of all, we would like to point out that the WAM-2layers model we used in this study is a moisture tracking model that numerically determines the moisture origin based on mass balance (see, e.g., van der Ent et al., 2013). We drew this conclusion mainly because, among all source regions with decreasing trends in annual and seasonal moisture contribution (Figs. 6 and 7, cf. Figs. 3 and 4), most are concentrated within SWS. In contrast, SS mainly experienced increasing trends in moisture contribution (although statistically insignificant).

Per your comment, we will further elaborate on the underlying mechanisms of this causal relationship in the revision. We will discuss this based on evaporation, vertical integrated moisture flux in the zonal direction, and vertical integrated moisture flux in the meridional direction based on reanalysis data. Figure R2 below shows the changes (trends) of these three fields in 1979–2015 and the two major pathways of moisture transport toward the target region (arrows in yellow). These two major pathways (from northern India Ocean to TGRR and from northwest Pacific Ocean to TGRR) are primarily based on Fig. 3, which have been identified in previous studies in Yangtze River basin (e.g., Xu et al., 2014). Note that the two key source regions, SWS and SS, are located along these two pathways and are critical to the moisture transport. Figure R3 shows 800-hPa, 500-hPa, and 300-hPa wind fields and their trends over time.

[Figure]

**Figure R2**: Trends of (a) evaporation, (b) vertical integrated moisture flux in the meridional direction (positive northward), and (c) vertical integrated moisture flux in the zonal direction (positive eastward) based on ERA-Interim during 1979–2015. Arrows in yellows show two major pathways of moisture transport toward the target region.

It is clear that the vast majority of all possible source regions experienced increased evaporation during the study period, despite a small portion of the SWS region with statistically insignificant decreased evaporation (Fig. R2a). Therefore, the enhanced evaporation increase over the SWS is unlikely the major cause of decreased precipitation in the TGRR. We then turn to vertical integrated moisture fluxes. As shown in Figs. 3 and R3, the two major pathways of moisture transport are controlled by winds at different pressure heights. The southwest pathway (from northern Indian Ocean) is mainly controlled by winds at relatively lower levels, while the southeast pathways (from northwest Pacific Ocean) is mainly controlled by winds at relatively higher levels. For the southwest pathway (from northern Indian Ocean), northward and eastward vertical integrated moisture fluxes in general enhanced along the pathway before reaching the SWS region (Fig. R2b and c). However, the further transport of moisture toward the TGRR is largely dampened by the decreased northward and eastward moisture flux over the eastern part of the SWS region, which contributes to the decreased precipitation in the TGRR. For the southeast pathway (from northwest Pacific Ocean), largely decreased eastward moisture flux over the northwest Pacific Ocean and South China Sea indicates an increased westward moisture contribution to the

target region. But this enhancement is partly offset by the decreased northward moisture flux along the pathway (especially over the SS region), which results in statistically insignificant trends as observed in Figs. 6 and 7.

[Figure]

**Figure R3**: Trends of 800-hpa (top), 500-hpa (middle), and 300-hpa (bottom) wind in the (a, b, and c) meridional direction (positive northward) and (d, e, and f) zonal direction (positive eastward) based on ERA-Interim during 1979–2015, overlaid with wind vectors (yellow arrows).

We will add the above analysis and Fig. R2 in the revision. Nevertheless, our understanding can be partly limited by the moisture tracking model as well as our selected reanalysis dataset used in this study. We will rely on more sophisticated models to investigate the dynamics of specific systems (e.g., monsoon system) for the same period in future study.

**Comment:** *The analysis of extremes has just 3 sample years each for wet and dry conditions, can the sample size be increased?*

**Response:** Thank you for the suggestion. We selected three sample years because these years are the wettest/driest over the TGRR. In fact, the selection of wet/dry years only has marginal impacts on the results, and will not alter our conclusions. To illustrate this, here we increase the sample size from three to five. The five wet years are 1982, 1983, 1989, 1998, 2007, and the five dry years are 1988, 1997, 2001, 2006, 2012. Results of moisture contribution and flux change are shown in

Fig. R4. It is clear that the patterns in Fig. R4 are very similar to those shown in Fig. 9 (3 sample years): extra moisture from the southwest regions during wet years, with weak negative signals in part of the adjacent regions southeast of the TGRR. Therefore, our conclusions are robust. We will add this figure in supplementary in the revision.

[Figure]

**Figure R4:** Difference in moisture (mm) contributing to annual (a) and summer (b) precipitation in the TGRR between five wet and five dry years (wet – dry), overlaid with the difference in vertical integrated moisture fluxes (blue arrows; wet – dry).

**Comment:** *The paper used the Water Accounting Model-2layers to simulate the moisture transport and to quantify and pinpoint the sources of moisture well, although the method has limitations it has been used quite effectively in this study.*

**Response:** Thanks again for all your comments and suggestions, which have substantially improved our manuscript.

**References**

van der Ent, R. J., Tuinenburg, O. A., Knoche, H. R., Kunstmann, H., & Savenije, H. H. G. (2013). Should we use a simple or complex model for moisture recycling and atmospheric moisture tracking?. Hydrology and Earth System Sciences, 17(12), 4869-4884.

Xu, X., Chen, L., Wang, X., Miao, Q., & Tao, S. (2004). Moisture transport source/sink structure of the Meiyu rain belt along the Yangtze River valley. Chinese Science Bulletin, 49(2), 181-188.

---

## Author Response (AR1)

**Response to Anonymous Referee #1**

**Comment:** This paper has addressed an important scientific question regarding the moisture sources contribution to the change in precipitation over the Three Gorges Reservoir Region. Title and abstract reflect the content of the paper clearly. This paper has outlined the details about the data, model and methodology clearly. My main concern is about the conclusions derived from the results. In general, this geographic region (TGRR) is influenced by the south-west monsoon (mainly in summer) which brings moisture from the Arabian Sea and Bay of Bengal, as is suggested by figures 3 and 4. Results also suggest (as is expected) the influence of westerlies system on the moisture source region. Then, the authors claim that the decreased precipitation over the TGRR during 1979-2015 is mainly due to the decreased moisture contribution from the source regions which span over the southwest of the TGRR, especially around the southeastern tip of the Tibetan Plateau.

**Response:** Thank you for your constructive comments and suggestions. Please see our responses below.

**Comment:** In order to establish the above-mentioned causation, authors presented trend analyses (figures 6, 7) and the time series (figure 8), which suggest the association between the SWS region and the TGRR. No physical mechanism is provided to explain how the decreasing moisture contribution from the SWS region (including the southeastern tip of the Tibetan Plateau) leads to the decreasing precipitation over the TGRR. In other words, it is not clear what is the pathway of this causation.

**Response:** Thank you for your comment and sorry for the confusion. First of all, we would like to point out that the WAM-2layers model we used in this study is a moisture tracking model that numerically determines the moisture origin based on mass balance (see, e.g., van der Ent et al., 2013). The causation is mainly reflected by the fact that among all source regions with decreasing trends in annual and seasonal moisture contribution (Figs. 6 and 7, cf. Figs. 3 and 4), most are concentrated within SWS. In contrast, SS mainly experienced increasing trends in moisture contribution (although statistically insignificant). This causation is physically-based instead of correlation-based. We intended to only use correlation coefficients to show the relationship between interannual variations of annual precipitation and moisture contributions. We have clarified this in the revised Section 3.2 (lines 252–273):

"We then define two key source regions based on the trends shown in Fig. 6: the southwestern source (SWS; 23°N to 29°N and 94°E to 104°E) and southern source (SS; 22°N to 29°N and 108°E to 112°E) regions. Figure 8 shows the interannual variations in the contributions of SWS, SS, WP, and EP to the change in TGRR annual precipitation (moisture averaged over the TGRR). For comparison purposes, the corresponding time series of local recycling (%) and TGRR precipitation (mm) are included. Note that both key regions (SWS and SS) experienced great changes (with the strongest trends; see Fig. 6) in moisture contribution, and are located along two major pathways that transport moisture to the YRB (from northern India Ocean and from northwest Pacific Ocean;

yellow arrows in Fig. 9) as identified in previous studies (Xu et al., 2004; Xu et al., 2019) and Fig. 3. Consistent with results in Fig. 6, the decreasing trends of annual moisture contribution from the WP (-60.33 mm/decade; p < 0.05) and SWS region (-9.16 mm/decade; p < 0.05) are much stronger than the insignificant weaking trend in EP (-2.37 mm/decade; p > 0.05) and the increasing trend in SS (1.45 mm/decade; p>0.05), dominating the precipitation decrease in the target region. The contribution of local recycling shows an increasing trend (0.07%/decade; p>0.05), indicating that moisture contribution from external sources on the annual scale is decreasing over time. This enhanced local recycling is in line with the increasing trend of reference evapotranspiration as observed in Lv et al. (2016) in 1982–2013. Seasonally for the two key source regions (Table 1), decreasing moisture contribution from the SWS region is observed in all seasons, and the decreasing trend in summer is stronger than the other three seasons, indicating the potential weakening contribution influenced by the Indian monsoon. We also evaluate the linear relationship between annual TGRR precipitation and moisture contributions of different source areas using the Pearson correlation coefficient (r). The r values for moisture contributions of WP, EP, SWS, SS, and local recycling are 0.69 (p < 0.05), 0.34 (p < 0.05), 0.68 (p < 0.05), 0.28 (p > 0.05), and -0.61 (p

**Figure R1**: Trends of (a) evaporation, (b) vertical integrated moisture flux in the meridional direction (positive northward), and (c) vertical integrated moisture flux in the zonal direction (positive eastward) based on ERA-Interim during 1979–2015. Arrows in yellows show two major pathways of moisture transport toward the target region.

---

## Referee Report (RR1)

Authors have added new figure, one table and the related description in the text. This concise version of the paper definitely clarifies my questions from the first round of revision. This version is written in a well-thought manner and is scientifically sound. I've only few comments and suggestions:

1. I understand why the authors have used 108°E as a line of reference to divide WP and EP of the domain. I would like to ask authors to add any relevant reference/s in the paragraph starting at line 197 to justify/support the use of this longitude. The discussion of longitude is also mentioned in the paragraph (starting line 225) on page number 8. As a result, it seems a bit arbitrary to use this longitude unless you clarify this clearly.

2. Line 258 : page 9: the phrase "great change" doesn't sound scientific. Please rewrite this sentence.

3. Line 295: This sentence has grammatical error. Please correct.

---

## Author Response (AR2)

**Response to Anonymous Referee #1**

**Comment:** *Authors have added new figure, one table and the related description in the text. This concise version of the paper definitely clarifies my questions from the first round of revision. This version is written in a well-thought manner and is scientifically sound. I've only few comments and suggestions:*

*1. I understand why the authors have used 108°E as a line of reference to divide WP and EP of the domain. I would like to ask authors to add any relevant reference/s in the paragraph starting at line 197 to justify/support the use of this longitude. The discussion of longitude is also mentioned in the paragraph (starting line 225) on page number 8. As a result, it seems a bit arbitrary to use this longitude unless you clarify this clearly.*

**Response:** Thank you for your comment. In this revision, we have added references in the paragraph staring from Line 196 to support the selection of 108°E (Xu et al., 2004; Zhou and Yu, 2005; Wei et al., 2012; Li et al., 2019b). To further clarify this, we have also added references (Xu et al., 2004; Wei et al., 2012) to support our discussion in the paragraph starting from Line 225 per your suggestion. Please see the revised Lines 199-200 and 238-239.

References:

Li, Y., Wu, L., Chen, X., and Zhou, W.: Impacts of Three Gorges Dam on regional circulation: A numerical simulation, J. Geophys. Res.-Atmos., 124(14), 7813–7824, https://doi.org/10.1029/2018jd029970, 2019b.

Wei, J., Dirmeyer, P. A., Bosilovich, M. G., and Wu, R.: Water vapor sources for Yangtze River Valley rainfall: Climatology, variability, and implications for rainfall forecasting, J. Geophys. Res., 117(D5), D05126, https://doi.org/10.1029/2011JD016902, 2012.

Xu, X., Chen, L., Wang, X., Miao, Q., and Tao, S.: Moisture transport source/sink structure of the Meiyu rain belt along the Yangtze River valley, Chin. Sci. Bull., 49(2), 181–188, https://doi.org/10.1360/03wd0047, 2004.

Zhou, T.-J., and Yu, R.-C.: Atmospheric water vapor transport associated with typical anomalous summer rainfall patterns in China, J. Geophys. Res., 110(D8), D08104, https://doi.org/10.1029/2004JD005413, 2005.

**Comment:** *2. Line 258: page 9: the phrase "great change" doesn't sound scientific. Please rewrite this sentence.*

We have changed "experienced great changes" to "experienced substantial changes" in the revised Line 259.

**Comment:** *3. Line 295: This sentence has grammatical error. Please correct.*

Thank you for pointing this out. We have revised this part to "In this section, we compare the

contributions of moisture sources during wet and dry years. We select three wet years (1989, 1998, and 2007) and three dry years (1997, 2001, and 2006) as the representative years from the annual precipitation time series (based on CMA observations; Fig. 2b)."